# The Role of Home-Based Exercise in Managing Common Musculoskeletal Disorders: A Narrative Review

**DOI:** 10.3390/jfmk10030326

**Published:** 2025-08-26

**Authors:** Vívian Santos Xavier Silva, Rodrigo José Battibugli Rivera, Eunice Fragoso Martins, Marco Carlos Uchida, Jean Marcos de Souza

**Affiliations:** 1School of Physical Education, Universidade Estadual de Campinas (UNICAMP), Campinas 13083-851, Brazil; v195476@dac.unicamp.br (V.S.X.S.); uchida@unicamp.br (M.C.U.); 2Department of Internal Medicine, School of Medical Sciences, Universidade Estadual de Campinas (UNICAMP), Campinas 13083-881, Brazil; e208215@dac.unicamp.br; 3Independent Researcher, Campinas 13083-881, Brazil; drigorivera@gmail.com; 4Post-Graduate Program in Medical Sciences, Universidade Estadual de Campinas (UNICAMP), Campinas 13083-887, Brazil

**Keywords:** musculoskeletal diseases, physical activity, home environment, low back pain, osteoarthritis

## Abstract

**Background**: Physical exercise can improve certain musculoskeletal conditions, but adherence remains low due to intimidating environments, limited government support, and financial constraints faced by many individuals. Home-based exercise is a potential strategy to treat that population. **Objective:** Discuss the main home-based resistance exercise protocols that have been studied and implemented for six highly prevalent musculoskeletal disorders. **Methods**: A narrative literature review was conducted, using the PubMed database to search for six highly prevalent musculoskeletal conditions: shoulder impingement syndrome (SIS), nonspecific low back pain (NSLBP), greater trochanteric pain syndrome (GTPS), knee osteoarthritis (OA), patellofemoral pain syndrome (PFPS), and Achilles tendinopathy (AT). The strategy included the listed pathologies and the keywords “physical exercise” or “physiotherapy”. Clinical trials, reviews, and retrospective studies from the last 30 years published in English, Portuguese, or Spanish were included. Only studies with sufficient details on the training protocols used and outcome measures were included in the analysis. **Results**: In SIS, exercise protocols have been more effective in the long term than in the short term. In PFPS and GTPS, strengthening the quadriceps and hip muscles helps reduce pain and improve function. For NSLBP, exercises like Pilates and core training demonstrate pain relief. In knee osteoarthritis, physical exercise improves pain, function, and quality of life both immediately and over time. Eccentric training promotes type I collagen formation in the tendons of patients with Achilles tendinopathy. **Conclusions**: Home-based resistance exercises studied and implemented in this review offer several general health benefits, including pain reduction, improved functionality, increased muscle strength, and enhanced motor control.

## 1. Introduction

The benefits of physical exercise for overall health have been extensively documented [1,2,3] and are routinely recommended in medical practice [2]. In recent decades, however, physical exercise has evolved from being a general health promotion recommendation to becoming a treatment tool for certain medical conditions, often ranking among the first intervention options. Numerous studies have evaluated the effectiveness of exercise for conditions such as fibromyalgia [4], mechanical low back pain [5], and osteoarthritis [6]. As research progresses, we may continue moving from generic exercise prescriptions for the general population to more disease-specific protocols tailored to each individual.

Verbal or written guidance from healthcare professionals regarding physical activity can increase patient adherence to exercise [7], but direct physical interventions are even more effective [8]. Despite the establishment of exercise centers as public policy in some countries and regions, participation remains low [9]. Some commonly cited reasons for the limited success of these initiatives include intimidating environments and inadequate supervision [9]. The situation is even more challenging for patients in lower-income countries, where government programs promoting exercise are scarce and patients often lack the resources to seek private physical therapists or exercise specialists.

Thus, home-based intervention models supported by the best available evidence could potentially improve adherence among the population. When considering science-based exercise prescription, however, while there is a vast number of papers analyzing the most efficient types of exercise for each condition, one may struggle to find well-described exercises to prescribe individually to their patients [10]. The goal of this literature review is to discuss the primary home-based resistance exercise protocols that have been studied and implemented for certain prevalent musculoskeletal disorders, where this form of exercise could offer significant treatment benefits. Whenever possible, we aimed to objectively list exercises that can be performed in an average household, with limited exercise equipment. Also, we sought to describe each exercise in detail in our Appendix B and Appendix A, hopefully providing guidance to health providers seeking home-based exercises for their patients.

## 2. Materials and Methods

We chose a narrative literature review as our strategy due to the broad scope of the topic. The PubMed database was used for our search, focusing on six highly prevalent musculoskeletal conditions: (1) shoulder impingement syndrome (SIS), (2) nonspecific low back pain (NSLBP), (3) greater trochanteric pain syndrome (GTPS), (4) patellofemoral pain syndrome (PFPS), (5) knee osteoarthritis (OA), and (6) Achilles tendinopathy (AT). The conditions were chosen based on the overall prevalence of musculoskeletal conditions [11], as well as our local epidemiology, specifically regarding situations in which a referral for physical rehabilitation was more likely [12]. We used the corresponding MeSH terms for each condition, along with the keywords “physical exercise” or “physical therapy”. We included clinical trials, reviews, and retrospective studies from the last 30 years, published in English, Portuguese, or Spanish. This approach, performed on 11 April 2024, yielded 13,792 results.

Initially, PubMed automation tools were used to select only full-text review articles, resulting in 2223 records. The remaining articles were reserved in a repository for future reference, if needed. Because the record list was still too long to screen, a second selection was performed using the PubMed “Best Match” tool, extracting only the first quartile of records for each musculoskeletal condition. This allowed the selection of 571 records for abstract screening. Researchers JMS and VSXS then selected studies that appeared relevant and had full-text versions available online. Duplicate, irrelevant, and case report studies were excluded, resulting in 85 studies in the final sample.

The studies that passed this preliminary selection were then fully reviewed by JMS, VSXS, RBR, and EFM. For the evidence synthesis, preference was given to systematic reviews and meta-analyses. For the exercise library, only studies providing sufficient detail on the training protocols and outcome measures were included, with preference given to clinical trials. A focus group composed of the same researchers met periodically to discuss the proposed exercises to ensure that the recommendations were feasible in a typical home environment in Brazil. The results of these discussions were compiled into the manuscript. A flowchart summarizing our methodology is provided in Figure 1.

## 3. Results and Discussion

As highlighted previously, despite the enormous amount of evidence regarding exercise as therapy for musculoskeletal conditions, finding specific home-based exercises was not easy. In particular, well-described exercises for NSLBP, the most prevalent musculoskeletal condition worldwide [11], were especially difficult to find, as noted by Pieri and colleagues [10]. Overall, however, we were able to synthesize a comprehensive list of well-described exercises for all the proposed conditions, which can be found within each subsection and in Appendix B and Appendix A, where instructions for each exercise are provided along with video footage. In Table 1, we also highlight the results of the clinical trials in which the exercises were implemented.

### 3.1. Shoulder Impingement Syndrome (SIS)

For patients with SIS, rehabilitation is the preferred treatment over surgery [27,28]. A systematic review by the Cochrane group summarizes various exercise protocols used in studies on rotator cuff diseases [29]. Overall, a meta-analysis was not possible due to heterogeneity in the data, but in the highest-quality trial, exercise was not superior to placebo in the short term; however, it showed better outcomes over time, suggesting that its effects are more long-term [13]. Ellenbecker and Cools also described a home-based exercise protocol specifically for patients with rotator cuff diseases [30]. Since a significant percentage of patients with SIS also present with tendinopathy [31], part of the rehabilitation protocol for both conditions overlaps, allowing for some extrapolation of exercises. Finally, Clausen and colleagues [14] outlined a more intense protocol specifically for SIS. Although the clinical trial for this protocol did not show differences compared with conventional treatment, both groups demonstrated improvements, and the training routine can be used as a progression for more advanced stages of rehabilitation.

In summary, while the evidence for resisted shoulder exercises in the treatment of SIS does not strongly favor them, they do not cause significant adverse effects [29] and may provide long-term benefits [13]. Additionally, given the overall health benefits of exercise, we believe it can be considered an adjunct therapy for pain management.

Below are examples of exercises mentioned in the studies discussed above that we believe can be performed in a home-based setting [13,14,29,30,32]:Progressive seated shoulder press, starting from a lying position with gradual changes in the angle of inclination until seated;Pendulum exercises;Wall push-ups;Wall slide exercises;Scapular plane elevation with extended arms and supinated hands;Internal rotation in a side-lying position;External rotation in a side-lying position;Standing scapular protraction–retraction;Abduction in a side-lying position;Prone shoulder extension;External rotation (both with the arm supported and free) in the scapular plane;Abduction in the scapular plane;Prone scapular retraction.

Regarding exercise volume, Ellenbecker and Cools suggest three sets of 15 to 20 repetitions [30], and most studies in the systematic review by Page and colleagues used two to three sessions per week [29]. However, due to the low intensity of these exercises, we believe that daily sessions are feasible.

### 3.2. Patellofemoral Pain Syndrome (PFPS)

PFPS is a common cause of knee pain, accounting for approximately 5% of anterior knee pain in patients under the age of 60 [33]. Biomechanical theories for its onset include patellar malalignment and joint overload [34]. When there is an imbalance in the tensile forces acting on the patella and the patellar tendon—whether due to relative tenomuscular dysfunctions (in the knee, feet, and/or hips) or overload—an abnormal displacement of the patella over the femoral condyle may occur, leading to inflammation [34]. However, it is important to recognize that micromolecular-level factors, such as osteometabolic activity of the patella and synovial inflammation, as well as local anatomical disorders, such as neuromas or soft tissue contractures, may also contribute to the condition, supporting a multifactorial view of the phenomenon [35,36].

The management of PFPS involves various approaches, including strengthening, stretching, biofeedback, orthotics, pharmacotherapy, and surgery [34,36,37]. In general, surgical therapy is not superior to conservative treatment [38,39]. Targeted muscle strengthening, especially of the quadriceps and hip muscles, has been widely studied [40]. Chiu and collaborators demonstrated that lower limb exercises increase knee strength and the contact area of the patellofemoral joint, reducing mechanical stress and pain [41]. Balci et al. suggest that squatting may be particularly effective [42]. Nakagawa, Fukuda, and their collaborators, in turn, reported better outcomes when quadriceps strengthening was combined with hip exercises [43,44], with possible benefits when the latter precedes the former [45].

In a recent meta-analysis evaluating pain outcomes for combined quadriceps and hip strengthening, despite substantial heterogeneity, analysis of three trials (n = 112) showed a significant reduction of 3.3 points on a 10-point scale compared with placebo, with sustained benefit at one year [40]. Furthermore, when combined hip and quadriceps strengthening was compared with isolated quadriceps strengthening, the former was superior [40]. Therefore, it is likely that combined strengthening is a more favorable strategy in the conservative management of PFPS.

Stretching may also provide benefits. Peeler and Anderson, for instance, reported pain improvement with static quadriceps stretching, although not strongly correlated with flexibility [46].

In practical terms, Greaves and collaborators developed a 6-week home-based rehabilitation program for PFPS, with specific guidelines [15]. In this trial, functionality improved following the intervention, with all participants pain-free at the end of the 6 weeks [15]. Another study, from 2016, focused specifically on runners, also provided a detailed description and progression of exercises [16]. This uncontrolled 8-week clinical trial showed improvements in pain and function, although it did not increase muscle strength [16].

In summary, the exercises from these two studies include [15,16]:Squat (to 90 degrees of knee flexion);Weighted squat (using a backpack);Squat with elastic band resistance for hip abduction;Squat with combined elastic band and external load;Unilateral squat (lunge);Supine glute bridge;Supine glute bridge with elastic band for hip abduction;Unilateral glute bridge;Unilateral glute bridge on an unstable surface;Unilateral glute bridge with elastic band and unstable surface;Lateral band walk;Seated knee extension with ankle weight;Side-lying hip abduction;Side-lying hip external rotation in flexion (clamshell) with elastic band;Step-up;Quadruped plank;Side plank with knee support;Single-leg squat;Plank;Side plank;Hamstring and calf stretch.

Considering training intensity, the protocol by Greaves and collaborators used a broad repetition range, from 10 to 25, with participants always aiming for 25. Resistance was provided using elastic bands, loaded backpacks, and ankle weights, corresponding to 5–25% of body weight. These were applied flexibly and individually according to perceived exertion [15]. In Esculier’s study, progressively more resistant elastic bands and increasingly higher steps were used [16], with sets of 10 to 15 repetitions or 10-second isometric holds, when applicable.

Regarding training volume, De Oliveira published a meta-analysis showing that programs with fewer sessions per week resulted in greater pain reduction [47]. It is possible that, due to the flexibility of these programs, adherence was higher in those groups. The authors therefore suggest that programs begin with 12 sets per session (e.g., 4 exercises with 3 sets each), two to three times per week.

### 3.3. Greater Trochanteric Pain Syndrome (GTPS)

Greater trochanteric pain syndrome (GTPS)—a clinical entity comprising gluteus medius tendinopathy, with or without associated trochanteric bursitis—is a common musculoskeletal disorder affecting the lateral region of the hip, with a variable prevalence ranging from 10% to 25% in developed countries [48,49].

Currently, conservative treatments for GTPS include physical modalities, exercise therapy, infiltrations, and lifestyle modifications. More specifically, these treatments may involve shockwave therapy, isolated eccentric training, slow high-intensity resistance training, corticosteroid injections, platelet-rich plasma injections, and dry needling [50]. In recent years, several meta-analyses have been published on conservative treatment approaches for GTPS [51,52,53]. However, due to limitations such as an insufficient number of relevant studies, heterogeneity in inclusion criteria, and a lack of comprehensive assessments across multiple outcomes (e.g., pain and function), the relative effectiveness of these treatments remains unclear. As such, no consensus has been reached on the ideal therapeutic approach [54]. For instance, a controlled trial by Ganderton and colleagues found that home-based, gluteal-focused exercise was no more effective than education—postural and work-related advice—combined with sham exercise [17]. These findings may suggest that poor habits are as important as gluteal strength in the development or prevention of tendinopathy.

Nevertheless, the most recent comprehensive network meta-analysis, conducted by Wang et al., reaffirmed the superiority of exercise therapy for treating GTPS. It demonstrated that exercise therapy led to the greatest reductions in pain (measured using the Numeric Rating Scale, NRS) and the most significant improvements in function (measured with the Victoria Institute of Sport Assessment–Gluteal questionnaire, VISA-G). This meta-analysis confirmed the effectiveness of exercise programs in reducing pain and enhancing functional outcomes in individuals with GTPS [54].

Based on the studies discussed, examples of home-based exercises investigated for GTPS include [17,18,19,20,55,56]:Piriformis/gluteal stretch;Supine bridge exercise (hip raise);Straight leg raises;Side-lying hip abduction with knees flexed;Single-leg bridge with hip abduction;Lateral walk with squat (right and left legs);Single-leg stance with contralateral hip flexion and extension;Wall squat;Iliotibial band stretch;Standing isometric hip abduction;Standing isotonic hip abduction slides;Lateral walk (right and left legs);Forward lunge (right and left legs);Side-lying isometric hip abduction;Single-leg wall squat;Lateral walk with squat (right and left legs);Unilateral bridge;Side-lying hip abduction (right and left legs);Hip adductor stretch (“butterfly stretch”);Trunk rotator stretch (right and left sides);Quadruped (all-fours) unilateral hip extension;Prone hip extension with 90° knee flexion;Transverse abdominis activation;Quadruped hip abduction (“fire hydrant”);

Regarding volume and intensity, the protocol proposed by Notarnicola et al. consisted of 30 min of bodyweight exercises per day, five days a week, over four weeks, totaling 20 sessions [18]. However, the number of repetitions was not reported. Clifford et al. implemented a protocol involving daily isometric and isotonic exercises, also performed five days per week, but over a 12-week period, totaling 60 sessions, with each session lasting six minutes [19]. The isometric arm included two exercises. In the first, a contraction of the gluteus medius was held for 30 s and repeated six times. In the second, which was weightbearing, the contraction was held for six seconds across 10 repetitions and three sets. The exercises in the isotonic arm were similar but performed for three sets of 10 repetitions [19]. In another study, patients were instructed to exercise every day, twice daily, for 12 weeks [17]. Exercises were performed for 5 to 15 repetitions over three or four sets [17]. Finally, the protocol by Nava et al. consisted of a motor control program carried out over eight weeks, with two weekly, in-person, individualized sessions. A total of 16 sessions were completed, each lasting 50 to 60 min. The protocol consisted of isotonic and isometric strengthening exercises focused on the hip abductor and extensor muscles, with coordination through verbal commands to improve dynamic motor control of the lower limbs [55]. Progression of the exercises occurred through elastic bands, moving from easiest to most difficult, and the addition of more challenging exercises [57].

Based on the protocols discussed, a reasonable volume of therapeutic exercise for individuals with GTPS would involve sessions performed five days per week over a period of 8 to 12 weeks, totaling approximately 60 sessions. The repetition target can be around 10 repetitions for three or four sets, aiming for approximately six minutes of time under tension per session. This training volume and intensity appear sufficient to promote pain reduction, functional improvement, and increased muscular strength in patients with GTPS.

### 3.4. Nonspecific Low Back Pain (NSLBP)

Nonspecific low back pain is a common condition—the most common among non-communicable diseases [58]—and a generally benign one, which can lead to various impairments in individuals who suffer from it, causing irregular movements that may exacerbate the pain [21]. As a consequence of these disturbances, disability has been analyzed as one of the major outcomes of the condition, being present worldwide and increasing over the past three decades [59]. Occupational activities are also affected, resulting in higher costs both in terms of medical care [59] and personal and professional issues due to work-related losses [60].

Low back pain is characterized as pain located below the last ribs and above the gluteal lines, which may or may not also affect the lower limbs [60]. Being a recurrent condition, it has several possible causes; however, the most prevalent form is nonspecific low back pain [60], meaning it has no specifically diagnosed cause and may relate to many factors.

In the search for better treatments for this condition, many researchers have focused on analyzing the relationship between physical exercise and nonspecific low back pain—the most prevalent form affecting individuals across all age groups [60]. A wide range of exercises has been described to support patients in staying active, engaging in physical activity, and returning to normal routines [61]. These include motor control exercises with low and high loads [21]; aerobic training; aquatic exercises; Pilates; resistance training; sling exercises (using slings and/or elastic bands); traditional Chinese exercises; walking; yoga [62]; and stretching, stabilization, coordination, balance, flexion and extension, and strengthening exercises [60]. Back schools have also been studied and offer a multidisciplinary approach involving education, workplace positioning, and re-education, along with physical exercises [63].

The most effective exercise model for treating chronic low back pain has not yet been clearly defined, which is why numerous studies—particularly systematic reviews—continue to explore this question. Despite the absence of a standardized approach, the various strategies mentioned above highlight potential tools that can be employed. However, many studies do not report their protocols transparently, making it difficult to reproduce the methods [10]. Details such as exercise intensity and volume are often lacking, requiring further measurement and planning before practical implementation, even when other documents are used as references.

Below are some examples of exercises utilized in trials [10,21,59,60,61]:Alternating Straight Leg Raises;Supine Lumbar Rotation Stretch (Spinal Twist Stretch);Double Knee-to-Chest Stretch (Hip and Knee Flexion);Supine Bridge with Shoulder Flexion (Arm Raises);Transverse Abdominis Isometric Activation (Abdominal Bracing);Glute Bridge Exercise;Isometric Hip Adduction with Pillow (Adductor Squeeze);Hip Abduction and Adduction with Knees Flexed (Side-Lying Hip Movements);Supine Isometric Foot Press Against Pillow;Standing Forward Trunk Flexion;Overhead Load Lift (High Load Lift from Overhead Position);Supine Straight Leg Raises;Cat–Cow Exercise (Quadruped Lumbar Flexion and Extension);Isometric Forearm Plank;Piriformis Stretch;Prone Alternate Arm and Leg Lifts (Swimming Exercise, Pilates Style);Bird-Dog Exercise (Quadruped Contralateral Arm and Leg Reach);Seated Lateral Flexion Stretch (Mermaid Stretch, Pilates).

Regarding intensity, most trials utilized a moderate repetition target, equivalent to 10 RM. A reasonable volume appears to be around two to three sets of three to five exercises, performed at least twice per week. For isometric exercises, the total time under tension per set can be 30 s. Most protocols were tested over a period of 8 to 12 weeks.

### 3.5. Knee Osteoarthritis (OA)

Knee osteoarthritis (OA) is a ubiquitous condition and one of the main reasons middle-aged and elderly people seek care in primary health. It is the 11th leading cause of disability worldwide [64] and affects around 10% of men over the age of 60 [65], with the prevalence in women probably twice as high [66]. Risk factors involved in the development of OA include genetic predisposition, aging, obesity, joint malalignment, and prior joint injury or surgery [67].

The pathogenesis of primary OA is only partially understood but involves disruption in the balance between the formation and degradation of cartilage [68]. Multiple factors contribute to this disruption, including excessive inflammatory mediators and matrix-degrading enzymes [68]. The predominant degeneration that follows prompts chondrocytes to clonally expand and increase their otherwise quiescent metabolism [69]. In older individuals, mitochondrial function can be impaired, and the increasing demands for metabolic activity can elevate reactive oxygen species (ROS), leading to a redox imbalance [70,71]. The overall oxidative stress leads to chondrocyte imbalance and apoptosis [72], which, in turn, eventually results in reduced production of the cartilage matrix and, thus, of the cartilage itself.

While the mechanisms described above account for the reduction in cartilage thickness and, therefore, in articular space, they do not explain the pain experienced by individuals with OA. Studies with volunteers undergoing arthroscopy without anesthesia have shown that cartilage is aneural and avascular, and therefore insensitive to touch or injury [73,74]. Local and central mechanisms of pain sensitization are likely involved in the manifestation of symptoms [74], as well as biodynamic factors [75].

Considering the multifactorial features of OA, it is tempting to consider physical exercise as a management option to address impaired muscle function [6]. A comprehensive Cochrane meta-analysis addressed this subject in 2015. Although the study found significant variability in the prescribed exercises, the pooled results—including aerobic and various forms of strengthening exercises—indicated that the interventions produced immediate effects on pain, physical function, and quality of life, lasting up to six months [6].

Regarding strengthening exercises in particular, the benefit was described more than two decades ago in the large FAST trial, in which resistive exercise over 18 months led to gains in physical function—measured by walking, lifting objects, and entering and exiting a car—knee pain—measured by the Knee Pain Scale [76]—and physical disability [77]. A recent network meta-analysis involving 2646 patients confirmed these findings: resistance training was superior to control in alleviating knee pain—measured by the Western Ontario and McMaster Universities Arthritis Index (WOMAC) pain domain—although it was not associated with improvements in the other dimensions of the WOMAC score [78].

As for home-based exercises, a meta-analysis by Si and collaborators, pooling results from 1442 patients, found that the home-based exercise arm improved significantly more than the control in both pain and function, as measured by the WOMAC index [79]. Although the meta-analysis found a significant bias related to blinding—since it is difficult to blind a home-based intervention—these results suggest that home-based exercises are a promising strategy to address knee OA [79].

Regarding the arsenal of home-based exercises, the following list provides some examples used in previous trials [22,23,24]:Seated knee extension (with cuffs or resistance bands);Knee extension with foam roll under the knee;Sit-to-stand;Step-ups;Forward touchdowns from a step;Partial wall squats;Side leg raises in standing;Crab walk with resistance band;Wall push while standing on the study leg;Bench knee curls (with cuffs or bands);Heel raises;Squats;Lunges;Glute bridge;Isometric quadriceps contractions;Lying straight-leg lifts;Prone leg lifts;Hip adductor isometric contraction;Tandem walk;Walking with dorsiflexed and plantarflexed ankle.

Regarding volume and intensity, most studies applied protocols lasting 6 to 12 weeks [22,23,24]. Session durations ranged from 20 to 60 min, three to five times per week [6,22,23,24]. The intensity was usually sufficient to achieve a 10-repetition maximum (RM) across 1 to 3 sets [6,23,24]. Therefore, a suggested exercise prescription for knee OA could consist of 12 weeks, with three sessions per week. Three sets of 3 to 5 exercises could be performed at a moderate intensity (around 10 RM), with 60 s of rest between sets. If isometric exercises are included, the time under tension per exercise can be around 50 s per set.

### 3.6. Achilles Tendinopathy (AT)

As inherent walkers, humans depend profoundly on the gastrocnemius and soleus muscles. The confluence of these two muscles forms a thick and resistant tendon—the Achilles tendon—which inserts into the calcaneus bone [80]. This structure is highly tensile and can reliably support around 10 times the body weight during running and 4 times during walking [81,82], making it the strongest tendon in the human body [83]. Nevertheless, it is prone to injury, accounting for 20% of all tendinopathies [84].

Tendinopathy occurs when matrix formation and organization become unbalanced relative to degradation [84]. Histologically, it is characterized by collagen fiber degeneration and disorganization, increased production of type III collagen and glycosaminoglycans, and decreased vascular ingrowth [85]. Nevertheless, tendinopathies generally do not exhibit overt inflammation [80].

Mechanisms leading to Achilles tendinopathy (AT) are multifactorial and relate to both intrinsic and extrinsic factors [84]. Intrinsic factors include age—older individuals [86]—gender—male [86,87]—tendon hypovascularity, and biomechanical abnormalities (e.g., forefoot varus and pes cavus) [80,84]. Examples of extrinsic factors include sudden increases in training, poor running technique, inappropriate footwear, or training on slippery surfaces [80,87,88].

Tackling the pain in AT is challenging because the mechanisms leading to symptoms are poorly understood. Inflammation is probably not the main pathological issue, as inflammatory infiltrate is not a hallmark of the condition [80,84]. On the other hand, the degree of tendon degeneration is not directly correlated with the amount of pain, so a purely structural explanation is also insufficient [89]. Some authors propose that leakage of extracellular matrix compounds (e.g., chondroitin sulfate) and lactate produced due to impaired tendon function may act as signaling agents, promoting the overproduction of pain mediators such as substance P and glutamate [89,90]. Therefore, measures to mitigate tendon dysfunction—such as physical exercise—should be central to the management of AT.

It has been demonstrated that eccentric exercise induces type I collagen formation in the tendons of patients with AT [91]. Accordingly, more homogeneous evidence regarding this type of training has emerged in recent years, making the body of evidence on AT rehabilitation less heterogeneous than that for the other pathologies discussed in this article. One of the most well-known eccentric high-load protocols for AT is the Alfredson protocol [90], discussed in detail further. In a meta-analysis by Wilson and collaborators, of the 22 articles included, the majority of the 19 trials investigating eccentric training used the Alfredson protocol [92].

Even before the publication of Alfredson’s protocol, however, Stanish and collaborators used a somewhat similar protocol, albeit with lower volume and including stretching and icing [93]. A clinical trial comparing the two interventions found that while both improved pain and function—measured by the Victorian Institute of Sport Assessment—Achilles (VISA-A)—Alfredson’s protocol led to greater improvement [26]. On the other hand, a 2014 study evaluating alternatives to the Alfredson protocol compared the classical approach to a modified one, in which the number of repetitions was limited by pain (as opposed to the “train through pain” directive of the high-volume protocol), and found no difference in VISA-A or visual analog pain scale outcomes [25]. In fact, a later meta-analysis also found no difference between Alfredson’s protocol and other heavy-load eccentric protocols with lower volume regarding pain outcomes [92].

While the relative homogeneity of rehabilitation strategies is beneficial for healthcare providers and patients, the monotony of the Alfredson protocol can limit adherence. To provide insights into the objective progression of exercises in AT rehabilitation, Baxter et al. analyzed 30 exercises for their kinetic and kinematic properties [94]. The authors created a composite loading index, allowing the exercises to be ranked according to the load imposed on the Achilles tendon. This enables healthcare professionals to offer a broader range of exercises adapted to the patient’s stage of recovery. However, it is worth noting that the hierarchy proposed by Baxter’s study has not yet been tested in clinical trials and should be applied with caution in clinical practice.

The Alfredson protocol (Figure 2): The protocol was developed and initially tested for mid-portion tendinopathy, where the disease—and consequently the point of maximal pain—is located in the middle of the tendon, approximately 2 to 6 cm proximal to its insertion at the calcaneus. Although some clinical trials evaluated the Alfredson protocol in unspecified AT (i.e., including both mid-portion and insertional forms), we suggest using the modified Alfredson protocol [95] for patients with insertional AT. In the original protocol, patients are instructed to perform 3 sets of 15 repetitions for two exercises—totaling 180 daily repetitions—twice daily, 7 days a week for 12 weeks. The first exercise is a unilateral eccentric heel drop from a step. No concentric phase should be performed—the healthy leg, the wall, or a handrail should be used to return to the starting position. The second exercise is similar but is performed with knees slightly bent, around 30 to 45 degrees. If the exercises are not sufficiently challenging, a backpack with weights should be used to bring the load to approximately 15 RM. Pain is expected and should not preclude training unless deemed disabling [90].

The modified Alfredson protocol (Figure 3): For insertional AT, a slight modification was made to the protocol to avoid loading during dorsiflexion [95]. The exercises are very similar, as are the intensity and volume. However, patients are instructed to perform the exercises on the floor rather than on a step. A slight lateral shift in the center of gravity towards the supporting leg is required during the concentric phase to position the affected leg for the eccentric movement. This modification limits the range of motion and avoids load during dorsiflexion.

Some additional examples from Baxter’s study [94] include, in ascending order of loading index:Seated heel raise;Step-up;Lunges;Jumping;Hopping.

## 4. Conclusions

Herein, we aimed to summarize and aggregate the current knowledge regarding the rehabilitation of six highly prevalent musculoskeletal conditions through home-based workouts. We listed the main exercises used in clinical trials and provided detailed instructions, along with video guides, in Appendix B and Appendix A. We conclude that the home-based resistance exercises studied here offer several general health benefits, including pain reduction, improved functionality, increased muscle strength, and enhanced motor control. Most exercises can be performed successfully with minimal equipment, such as elastic bands or ankle weights. When even these are unavailable, items like a loaded backpack, a chair, or a staircase can serve as substitutes, making these routines adaptable to most households. The ability to exercise at home may be a game-changer in physical rehabilitation, and future studies exploring these approaches are likely to be well received.

## Figures and Tables

**Figure 1 jfmk-10-00326-f001:**
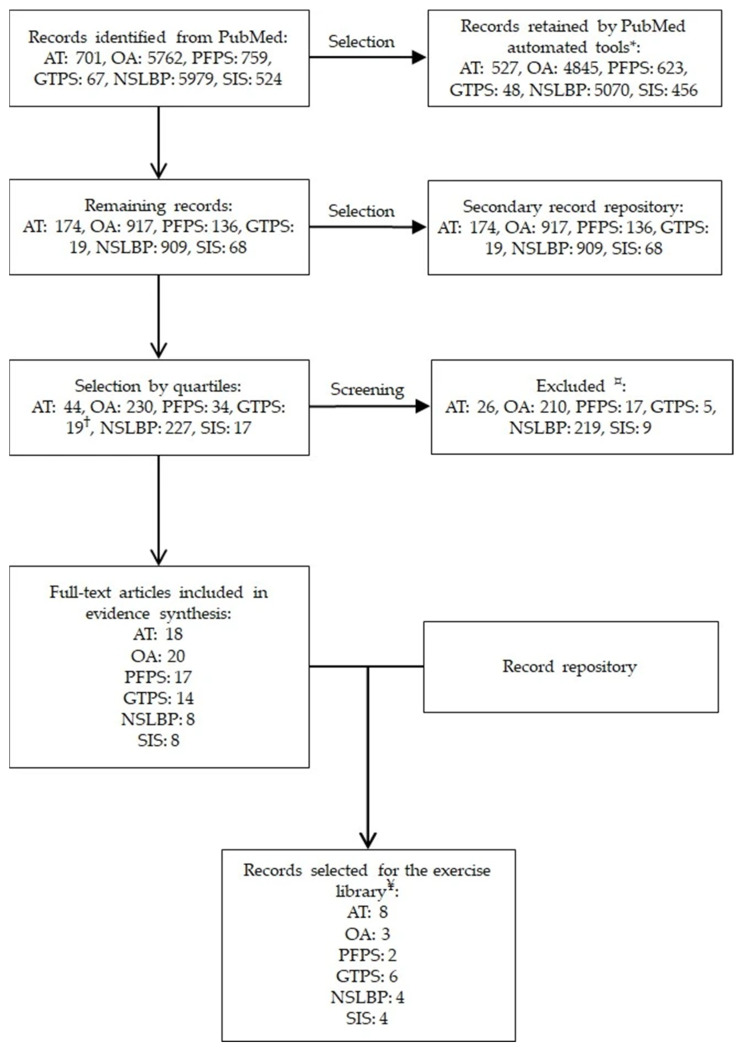
Flowchart illustrating the study search and selection protocol. * PubMed automation tools were used to select full-text reviews only. The remaining records were retained as first-level repository. † Because greater trochanteric pain syndrome’s search strategy only yielded 19 results, quartile selection was not performed. ¤ Duplicate, irrelevant, and case report studies were definitely excluded from the study. ¥ Only studies implementing fully home-based exercises with adequate descriptions were included in the exercise library. Records from the nested repository were used to compile these lists. Abbreviations: AT: Achilles tendinopathy; OA: Knee osteoarthritis; PFPS: Patellofemoral pain syndrome; GTPS: greater trochanteric pain syndrome; NSLBP: nonspecific low back pain; SIS: shoulder impingement syndrome.

**Figure 2 jfmk-10-00326-f002:**
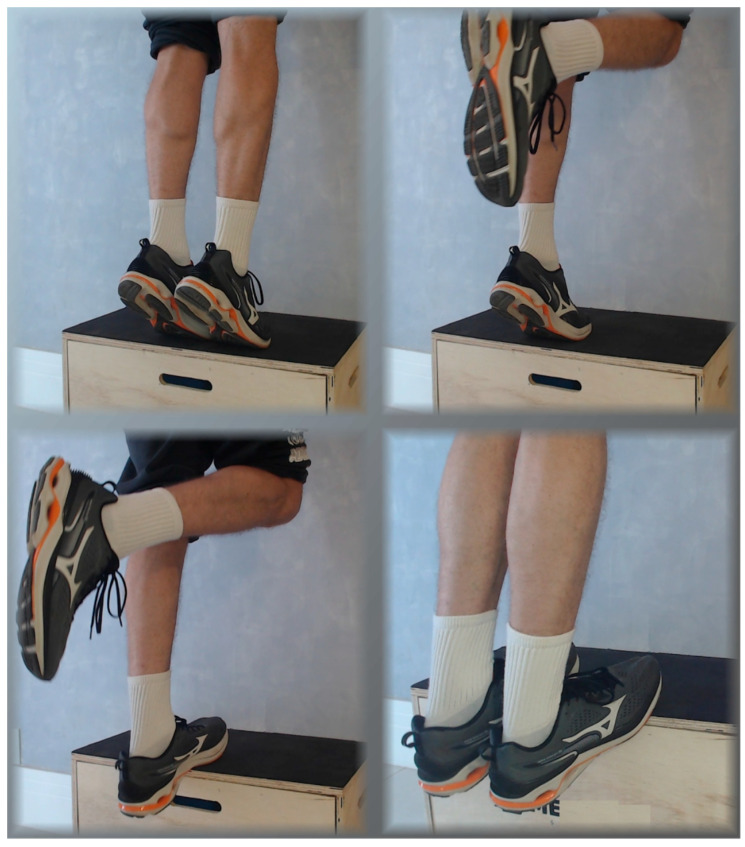
The photographs illustrate the progression of a unilateral eccentric heel drop exercise, recommended for mid-portion Achilles tendinopathy. Standing on a step or jump box, the individual uses both legs—or a handrail—to rise into maximal bilateral plantar flexion. Once at the top, the non-working leg is lifted, and the working leg slowly lowers the body in a controlled eccentric motion until reaching maximal dorsiflexion. The non-working leg then assists by joining the working leg to help lift the body back to the starting position. This ensures the exercise emphasizes the eccentric (lowering) phase. The second exercise in the protocol follows the same pattern but is performed with the knees slightly bent.

**Figure 3 jfmk-10-00326-f003:**
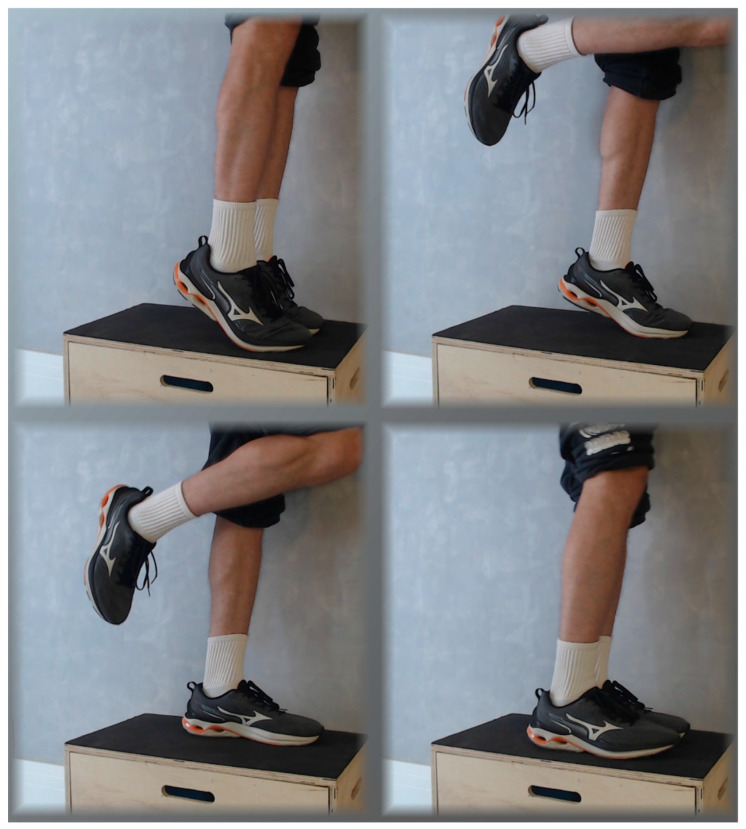
The photographs depict the progressions for a partial unilateral eccentric heel drop performed on the floor, suggested for insertional Achilles tendinopathy. The limitation of the dorsiflexion angle imposed by the floor prevents maximal stretching of the tendon. The individual uses both legs to lift into bilateral maximal plantar flexion, placing more effort on the healthy leg. From there, the rest leg is suspended and the working leg slowly drops in an eccentric motion until reaching the floor. Then, the rest leg once again joins the working leg to lift the whole body back to maximal plantar flexion, starting the exercise again. The second exercise of the protocol is performed in exactly the same way, but with the knees slightly bent.

**Table 1 jfmk-10-00326-t001:** Summary of findings of the clinical trials utilized for the exercise library.

Condition	First Author, Year	N	Intervention	Control	Outcomes of Interest	Results	Observations
SIS	Bennell, 2010 [13]	120	Manual therapy (10w) plus home exercises (12w)	“Sham” therapy, no exercise	SPADI	NS	
					NRS	Favors active	
					Global rate	Favors active	
					SF36	Favors active	
SIS	Clausen, 2021 [14]	200	Home-based high-volume training, 15w	Standard care	SPADI	NS	All better within groups
					Shoulder strength	NS	
					NRS	NS	
PFPS	Greaves, 2021 [15]	16	Home-based resistance training, 6w	N/A	NRS	NS	
					KOOS	↑	
					AKPS	↑	
					Quadriceps strength	NS	
					TSK	↑	
PFPS	Esculier, 2016 [16]	21	Home-based resistance training, 8w	N/A	KOS-ADLS	↑	
					VAS-U	↑	
					VAS-W	↑	
					VAS-R	↑	
					Strength	NS	
GTPS	Ganderton, 2018 [17]	94	Home-based resistance training, 12w	Sham-exercise	VISA-G	NS	All better within groups. A subanalysis of responders showed the active arm responded better
					OHS	NS	
					AQoL	NS	
					HOOS	NS	
GTPS	Notarnicola, 2023 [18]	44	Home-based resistance training, 4w	ESWT, crossover trial	NRS	NS	All better within groups
					LEFS	NS	
					RMS	NS	
GTPS	Clifford, 2019 [19]	30	Home-based isometric training, 12w	Home-based isotonic training, 12w	VISA-G	NS	Effect size better for isotonic, but no difference within or between groups
					NRS	NS	
					GRC	NS	
					PCS	NS	
					HOOS	NS	Pain and QoL domains of HOOS within group difference in isotonic
					EuroQoL 5D-5L	NS	
GTPS	Marcioli, 2024 [20]	26	Bodyweight training, hips and core, 4w	Bodyweight training, hips only, 4w	VAS	NS	Improvements within groups for VISA and VAS
					VISA-G	NS	
					PBT	NS	
					SBT	↑	
NSLBP	Aasa, 2015 [21]	70	Bodyweight resistance training, 8w	High-load, gym-based training, 8w	PSFS	↑	↑PSFS sustained for 12mo
					VAS	NS	
					PPB	Mostly NS	All better within groups
OA	Bennell, 2017 [22]	148	Home-based resistance training, 3mo	Education only	NRS	↑	
					WOMAC	↑	
					AQoL-2	↑	
OA	Chen, 2019 [23]	141	Home-based resistance training, 12w	Education only	WOMAC	↑	
					STS	↑	
					TUG	↑	
					AIMS2-SF	↑	
OA	Kuru Çolak, 2017 [24]	78	Home-based resistance training, 6w	PT	VAS	PT better	
					6MWT	NS	
					Quadriceps strength	PT better	
AT	Stevens, 2014 [25]	28	Alfredson protocol, “do as tolerated” volume, 6w	Alfredson protocol, standard, 6w	VISA-A	NS	
					VAS	NS	
AT	Stasinopoulos, 2013 [26]	41	Alfredson’s protocol	Stanish’s protocol	VISA-A	Superiority of Alfredson’s protocol	Both improved

↑ Refers to improvement in the respective scale. 6MWT: Six-minute Walking Test; AIMS2-SF: Short form of the Arthritis Impact Measurement Scales 2; AKPS: Kujala Anterior Knee Pain Scale; AQoL: Assessment of Quality of Life; AT: Achilles tendinopathy; ESWT: Extracorporeal Shockwave Therapy; GRC: Global rating of change; GTPS: Greater trochanteric pain syndrome; HOOS: Hip Disability and Osteoarthritis Outcome Score; KOOS: Knee Injury and Osteoarthritis Outcome Score; KOS-ADLS: Knee Outcome Survey—Activities of Daily Living Scale; LEFS: Lower Extremity Functional Scale; NRS: Numeric rating scale; NS: non-significant; NSLBP: Nonspecific low back pain; OA: Knee osteoarthritis; OHS: Oxford Hip Score; PBT: Prone Bridge Test; PCS: Pain Catastrophizing Scale: PFPS: Patellofemoral pain syndrome; PPB: Physical Performance Battery; PSFS: Patient-Specific Functional Scale; PT: Physiotherapy; QoL: Quality of life; RMS: Roles and Maudslay Score; SBT: Supine Bridge Test; SF36: 36-item Short Form Health Survey; SIS: Shoulder impingement syndrome; SPADI: Shoulder Pain and Disability Index; STS: Sit to stand test; TSK: Tampa Scale for Kinesiophobia; TUG: Timed up-and-go test; VAS-R: Visual analogue scale—Running pain; VAS-U: Visual analogue scale—Usual pain; VAS-W: Visual analogue scale—Worst pain; VISA-A: Victorian Institute of Sports Assessment—Achilles tendinopathy; VISA-G: Victorian Institute of Sports Assessment—Gluteal Tendinopathy; WOMAC: Western Ontario McMaster Universities Osteoarthritis Index.

## Data Availability

All gathered data are available in this manuscript.

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
