# Peer review of "The Role of Home-Based Exercise in Managing Common Musculoskeletal Disorders: A Narrative Review"

_jfmk, 2025, doi:10.3390/jfmk10030326_

Round 1

Reviewer 1 Report

Comments and Suggestions for Authors

Dear Authors

The manuscript entitled “The Role of Home-Based Exercise in Managing Common
Musculoskeletal Disorders: A Narrative Review” is presented as a work based on a narrative review, and is relevant from the point of view of the synthesis carried out on the subject, namely in relation to the most prevalent musculoskeletal conditions.

The subject is pertinent, since it focuses on the benefits of resistance exercise in improving pain, functionality, strength and motor control, in the various pathologies addressed. It is, however, as the authors recognize, a widely studied subject, including through systematic reviews. In this sense, the authors should justify the option of a narrative review, especially since the conclusions are not new, they merely confirm the results of previous studies.

From a methodological point of view, a narrative review allows for greater flexibility in design, so that the review is generally adequate, but some aspects could be improved:

It is advisable to include a Flowchart of empirical studies and reviews, which could make the description of the review more clear and objective.

Regarding the results and discussion, despite the decision to include studies from the last 30 years, in which systematic reviews were performed, the synthesis carried out is acceptable and interesting.

However, it would be interesting to carry out a more detailed analysis of the lists of exercises proposed for each of the pathologies, namely, identifying and prioritizing those that could produce more and better benefits in terms of each of the parameters under consideration (pain, functionality, strength, etc.). I believe that in this way the authors would add some originality to the work.

Regarding the conclusions, although they generally reflect the objective, they highlight the little contribution to increasing knowledge on the subject.

In this sense, perhaps highlighting the aggregation of knowledge on the most prevalent musculoskeletal conditions.

I also emphasize, in relation to bibliographic references, that only a quarter of them are after 2020. This reinforces my previous comment about a better justification for opting for this narrative review.

Author Response

Dear Reviewers and Editors,

We are truly grateful for the time and effort you dedicated to reviewing our manuscript. Your suggestions have been very insightful and have certainly improved our work.

Please find our replies below.

Best regards.

--Reviewer 1--

Dear Authors

The manuscript entitled “The Role of Home-Based Exercise in Managing Common Musculoskeletal Disorders: A Narrative Review” is presented as a work based on a narrative review, and is relevant from the point of view of the synthesis carried out on the subject, namely in relation to the most prevalent musculoskeletal conditions. The subject is pertinent, since it focuses on the benefits of resistance exercise in improving pain, functionality, strength and motor control, in the various pathologies addressed. It is, however, as the authors recognize, a widely studied subject, including through systematic reviews. In this sense, the authors should justify the option of a narrative review, especially since the conclusions are not new, they merely confirm the results of previous studies.

Reply: Thank you for the recognition. When planning this project, our goal was not to produce new evidence—as would be expected in a typical meta-analysis—but to aggregate the widely scattered information in an accessible and didactic way, creating a sort of clinical handbook for prescribing exercises, without losing the best available evidence approach. To clarify this, we added a new sentence at the end of the introduction, as follows:

Also, we sought to describe each exercise in detail in our Appendix and Supplementary files, hopefully providing guidance to health providers seeking home-based exercises for their patients.

In addition, following the reviewers’ advice, we have included a comprehensive explanation of each exercise, along with video descriptions. These can be found in our Appendix and Supplementary files.

From a methodological point of view, a narrative review allows for greater flexibility in design, so that the review is generally adequate, but some aspects could be improved: It is advisable to include a Flowchart of empirical studies and reviews, which could make the description of the review more clear and objective.

Reply: You are right. We have added a flowchart describing the search results and included studies. In the methodology, the search strategy and selection of studies is also more detailed (pages 2 and 3). Please refer to Figure 1.

Regarding the results and discussion, despite the decision to include studies from the last 30 years, in which systematic reviews were performed, the synthesis carried out is acceptable and interesting. However, it would be interesting to carry out a more detailed analysis of the lists of exercises proposed for each of the pathologies, namely, identifying and prioritizing those that could produce more and better benefits in terms of each of the parameters under consideration (pain, functionality, strength, etc.). I believe that in this way the authors would add some originality to the work.

Reply: Thank you for the highlight. We have now provided a table summarizing the main outcomes of the studies used for the exercise library. This allows the reader to grasp the specific benefits of each exercise for every musculoskeletal condition. Please refer to Table 1.

Regarding the conclusions, although they generally reflect the objective, they highlight the little contribution to increasing knowledge on the subject. In this sense, perhaps highlighting the aggregation of knowledge on the most prevalent musculoskeletal conditions.

Reply: We believe the issue has now been addressed in the previous reply. Please refer to Table 1. Also, the conclusion was reformulated, aiming to reinforce our main goals with the manuscript:

“Herein, we aimed to summarize and aggregate the current knowledge regarding the rehabilitation of six highly prevalent musculoskeletal conditions through home-based workouts. We listed the main exercises used in clinical trials and provided detailed instructions, along with video guides, in the Appendix and Supplementary Materials.”

I also emphasize, in relation to bibliographic references, that only a quarter of them are after 2020. This reinforces my previous comment about a better justification for opting for this narrative review.

Reply: We understand your emphasis. But the objective of the manuscript was something entirely different, as highlighted in our previous replies. We hope one can understand our goal now.

Thank you again.

Reviewer 2 Report

Comments and Suggestions for Authors

I appreciate the opportunity to review this manuscript. It offers an interesting narrative review on the effects of home-based resistance exercise protocols that have been studied and implemented for certain prevalent musculoskeletal disorders.  

The review is well-structured and overall well-written, offering an extensive literature overview. However, the results are not effectively presented. The listed exercises must be explained, or illustrated with a photograph, in order that this document could serve as valuable reference material for students, educators, researchers, and clinicians.

My proposed corrections are major, as follows.

General comments

Please, add one space between the word and the reference number, according to the recommendations of the Journal.

One of the aims of the study is the next: “to objectively list exercises that can be performed in an average household, with limited exercise equipment”. Please, enhance and improve the explanation for each of the exercises proposed in the revision. As listed, the reader must consult the original source to learn how to do the exercise, so the revision fails to fulfill its purpose and should not be published.

Specific comments

Line 70. “The PubMed database was used for our search, focusing on six highly prevalent musculoskeletal conditions”. I think that you have to add a justification on the selection of these six prevalent musculoskeletal conditions.

Line 82. Please, add the total number of screened studies for each of the selected conditions.

Line 94. “Ellebecker and Cools also described a home-based exercise protocol specifically for patients with similar conditions[15].” Please, clarify this sentence. It has no meaning in its context.

Line 190: “Resistance was provided using elastic bands, loaded backpacks, and ankle weights, producing loads equivalent to 5% to 25% of body weight, applied flexibly and individually according to perceived exertion.” Please, syntax review.

Line 199. Please, add the number of sessions/week.

Line 504. “A slight lateral shift in the center of gravity is required during the concentric phase to position the affected leg for the eccentric movement. This modification limits the range of motion and avoids load during dorsiflexion.” Please, clarify to which side it is necessary to make the lateral shift.

Line 531. “We conclude that the home-based resistance exercises studied and implemented here offer several general health benefits, including pain reduction, improved functionality, increased muscle strength, and enhanced motor control.” Please, correct the previous sentence. You have not implemented the resistance exercises you have studied and showed them.

Author Response

Dear Reviewers and Editors,

We are truly grateful for the time and effort you dedicated to reviewing our manuscript. Your suggestions have been very insightful and have certainly improved our work.

Please find our replies below.

Best regards.

--Reviewer 2--

I appreciate the opportunity to review this manuscript. It offers an interesting narrative review on the effects of home-based resistance exercise protocols that have been studied and implemented for certain prevalent musculoskeletal disorders.  

The review is well-structured and overall well-written, offering an extensive literature overview. However, the results are not effectively presented. The listed exercises must be explained, or illustrated with a photograph, in order that this document could serve as valuable reference material for students, educators, researchers, and clinicians.

Reply: We appreciate the recognition. You are absolutely correct about the vagueness of the lists provided. As it would be too long to describe every exercise within the manuscript, we added an Appendix and Supplementary file with not only the descriptions of the exercises but also video footage of the suggested execution.

Please, add one space between the word and the reference number, according to the recommendations of the Journal.

Reply: Ok. They were corrected.

One of the aims of the study is the next: “to objectively list exercises that can be performed in an average household, with limited exercise equipment”. Please, enhance and improve the explanation for each of the exercises proposed in the revision. As listed, the reader must consult the original source to learn how to do the exercise, so the revision fails to fulfill its purpose and should not be published.

Reply: Sure. Please, refer to the first reply and to the Appendix/Supplementary Material.

Line 70. “The PubMed database was used for our search, focusing on six highly prevalent musculoskeletal conditions”. I think that you have to add a justification on the selection of these six prevalent musculoskeletal conditions.

Reply: We agree. Our choice was based on our local epidemiology. We have now clarified this in the corrected methodology section:

“The conditions were chosen based on the overall prevalence of musculoskeletal condi-tions [11], as well as our local epidemiology, specifically regarding situations in which a referral for physical rehabilitation was more likely [12].

Line 82. Please, add the total number of screened studies for each of the selected conditions.

Reply: Sure. We have added a more comprehensive description of the search strategy and selection stages. Also, Figure 1 now summarizes it.

“Initially, PubMed automation tools were used to select only full-text review articles, resulting in 2,223 records. The remaining articles were reserved in a repository for future reference, if needed. Because the record list was still too long to screen, a second selection was performed using the PubMed “Best Match” tool, extracting only the first quartile of records for each musculoskeletal condition. This allowed the selection of 571 records for abstract screening. Researchers JMS and VSXS then selected studies that appeared relevant and had full-text versions available online. Duplicate, irrelevant, and case report studies were excluded, resulting in 85 studies in the final sample.”

Line 94. “Ellebecker and Cools also described a home-based exercise protocol specifically for patients with similar conditions[15].” Please, clarify this sentence. It has no meaning in its context.

Reply: We apologize for the imprecision. We were referring to the possibility to extrapolate exercises for rotator cuff disease sensu lato to shoulder impingement syndrome. The text now reads:

“Ellenbecker and Cools also described a home-based exercise protocol specifically for patients with rotator cuff diseases.”

Line 190: “Resistance was provided using elastic bands, loaded backpacks, and ankle weights, producing loads equivalent to 5% to 25% of body weight, applied flexibly and individually according to perceived exertion.” Please, syntax review.

Reply: You are right. We have rephrased it.

“Resistance was provided using elastic bands, loaded backpacks, and ankle weights, corresponding to 5%–25% of body weight. These were applied flexibly and individually according to perceived exertion.”

Line 199. Please, add the number of sessions/week.

Reply: Thank you for pointing that out. It was a valuable missing information. We have added it:

“The authors therefore suggest that programs begin with 12 sets per session (e.g., 4 exercises with 3 sets each), 2 to 3 times per week.”

Line 504. “A slight lateral shift in the center of gravity is required during the concentric phase to position the affected leg for the eccentric movement. This modification limits the range of motion and avoids load during dorsiflexion.” Please, clarify to which side it is necessary to make the lateral shift.

Reply: Sure. It now reads:

“A slight lateral shift in the center of gravity towards the supporting leg is required during the concentric phase to position the affected leg for the eccentric movement.”

Line 531. “We conclude that the home-based resistance exercises studied and implemented here offer several general health benefits, including pain reduction, improved functionality, increased muscle strength, and enhanced motor control.” Please, correct the previous sentence. You have not implemented the resistance exercises you have studied and showed them.

Reply: Of course. You are absolutely correct. Now it reads:

“We conclude that the home-based resistance exercises studied here offer several general health benefits (…)”

Thank you again!

Round 2

Reviewer 2 Report

Comments and Suggestions for Authors

I want to thank the authors for the implementation of all the corrections proposed. Congratulations on the publication of your manuscript.